# The clinical efficacy and safety of equipment-assisted intravesical instillation of mitomycin C after transurethral resection of bladder tumour in patients with nonmuscular invasive bladder cancer: A meta-analysis

Weijian Zhou[1], Jianping Liu[2], Dongdong Mao[3], Changying Hu[4], Dianjun Gao[1]*

1 Department of Clinical Medicine, Weifang Medical University, Weifang, China, 2 Department of Urology, The First People's Hospital, Yibin, China, 3 Department of Urology, Affiliated Hospital of Weifang Medical University, Weifang, China, 4 Department of Clinical Medicine, The Second Affiliated Hospital of Chengdu Medical College Nuclear Industry 416 Hospital, Chengdu, China

* sdwfgdj@163.com

## Abstract

### Background

This review and meta-analysis aimed to systematically evaluate the clinical efficacy and safety of equipment-assisted intravesical instillation of mitomycin C (MMC) in patients with nonmuscular invasive bladder cancer (NMIBC) after transurethral resection of bladder tumour (TURBT).

### Methods

The Embase, PubMed, CNKI, CBM, WANGFANG, VIP, Cochrane Library, and Clinicaltrial. com databases were searched for articles published before April 2022. The experimental group was treated with intravesical instillation of MMC assisted by equipment, including radiofrequency-induced thermochemotherapy, conductive thermochemical therapy, electromotive drug administration, or locoregional hyperthermia. The control group was treated with simple MMC perfusion. The outcomes of interest in the meta-analysis were recurrence, progression, side-effects, gross haematuria, and bladder irritation.

### Results

A total of 15 studies that enrolled 1,190 patients were included in the meta-analysis. Compared to that of the control group, device-assisted intravesical instillation of MMC significantly reduced both tumour recurrence (odds ratio [OR] = 0.32, 95% confidence interval [CI] [0.24, 0.42], P <0.00001) and progression (OR = 0.29, 95% CI [0.12, 0.67], P = 0.004). There were no significant differences between the two groups in terms of safety (OR = 1.21, 95% CI [0.66,2.21], P = 0.54), bladder irritation (OR = 1.06, 95% CI [0.72,1.55], P = 0.78), or gross haematuria (OR = 1.11, 95% CI [0.64,1.94], P = 0.72).

**Data Availability Statement:** All relevant data are within the manuscript and its Supporting Information files.

**Funding:** The author(s) received no specific funding for this work.

**Competing interests:** The authors have declared that no competing interests exist.

## Conclusions

Equipment-assisted intravesical instillation of MMC significantly reduced the recurrence and progression of patients with NMIBC who underwent TURBT and improved their quality of life. Given the significant heterogeneity in research quality and sample size among earlier studies, more prospective, multicentre, large sample randomized controlled trials are needed to supplement and verify this in the future.

## Introduction

Bladder cancer is the ninth most common malignancy in the world [1]. According to histopathological staging, 90% of the cases are uroepithelial carcinoma (UC), and about 75% of the diagnosed UC cases are nonmuscle invasive [2]; in turn, 70% of the nonmuscle invasive UC cases are advanced stage or associated with metastasis. Patients with UC are typically treated with palliative care and have a 5-year survival rate of only 15% [3]. According to the severity of the disease and the degree of invasion, bladder cancer is roughly divided into muscular invasive and nonmuscular invasive carcinoma (NMIBC) [4]. More than 75% of the cases are diagnosed as NMIBC [5]. Currently, the mainstay of treatment for NMIBC is transurethral resection of bladder tumour (TURBT) [6]. The postoperative local recurrence rate (42–80%) and the progression rate (2–45%) are high. Therefore, multiple repeated transurethral resections, intravesical chemotherapy, or intravesical immunotherapy are often needed after the initial surgery [7]. For patients with NMIBC who are at a high risk of recurrence, mitomycin C (MMC) or Basile Calmette–Guerin (BCG) perfusion therapy is the main form of treatment [8]. However, there is a shortage of available BCG products [9]. Besides, BCG is likely to cause severe local and systemic side-effects in patients with Ta or T1 bladder cancer [10]. The optimal dose and course of treatment when MMC is used as an adjuvant infusion therapy remain controversial. The 3–5 year recurrence rate in patients treated with MMC is only 8% lower than that of patients treated with TURBT [8]. Therefore, it is particularly urgent and important to improve the perfusion efficiency of MMC. Some studies have described the intravesical instillation therapy of MMC assisted by equipment, including radiofrequency induced thermochemotherapy (RITE), conductive thermochemical therapy (CHT), electromotive drug administration (EMDA), and locoregional hyperthermia (LRH) [11]. Most of these results show the superiority of device-assisted MMC perfusion therapy. However, this meta-analysis was designed to address current issues posed by device variability among studies and the small sample sizes of individual studies. Comparing the effectiveness and safety of device-assisted MMC bladder perfusion against MMC bladder perfusion alone provides an evidence-based medical reference to optimize clinical treatment protocols for patients with NMIBC after TURBT as well as a more reliable and efficient clinical option for both patients and physicians.

## Methods

This meta-analysis was registered a priori with the International Prospective Registration of Systematic Reviews (PROSPERO) on 30 May, 2022 (registration number: CRD42022333967, PROSPERO (york.ac.uk)). This meta-analysis was based on the Preferred Reporting Items for Systematic Review and Meta-Analyses guidelines [12], which can be used to assess the conformance of our results (**S1 File**).

## Search strategy

Two researchers searched eight electronic databases (Embase, Pubmed, CNKI, CBM, WANG-FANG, VIP, Cochrane Library, and Clinicaltrial.com) with relevant subject words and corresponding free words until April 2022. The detailed search process is described in **S2 File**.

## Selection criteria

Studies that met the PICOS criteria were included in the meta-analysis: The selection criteria were as follows: 1) randomized controlled trials (RCT); 2) studies that enrolled patients with pathologically confirmed noninvasive bladder cancer treated with TURBT; 3) studies that included an experimental group treated with equipment-assisted intravesical MMC instillation (RITE, CHT, EMDA, or LRH) and the control group was treated with simple MMC infusion; 4) recurrence rate, progression rate, safety indicators, including side-effect profiles (the ratio of the number of people with side-effects after intervention to the total number of people), gross haematuria, and bladder irritation sign. The exclusion criteria were as follows: 1) duplicate publications; 2) nonrandomized controlled trials; 3) studies that enrolled subjects with other significant comorbidites; 4) incomplete or biased results; 5) multiple drugs combined or continuous treatment.

## Data extraction

Two researchers independently screened the literature and extracted study data according to the title and abstract. The full-text publications that met the inclusion criteria were searched. A third researcher resolved any discrepancies until a consensus was reached. Standardized tables were used to extract data, as summarized in **Table 1**. The measures of interest were as follows: the name of the first author, the year of publication, the follow-up time of the patient, the average age of the sample, the proportion of male participants, intervention measures, and study outcomes (i.e., recurrence rates, progression rates, side-effects, gross haematuria, and bladder irritation).

## Quality assessment

Two researchers used the Cochrane Collaboration Reviewer's Handbook and the Cochrane Risk of Bias (ROB) tool to evaluate the quality of the studies included in the meta-analysis. Any differences were resolved through discussion and consensus between the two researchers. The ROB tool evaluates research quality in seven areas, including random sequence generation, allocation hiding, blindness of participants and personnel, blindness of outcome evaluation, incomplete outcome data, selective reporting, and other potential sources of bias.

## Statistical analysis

Revman 5.3 software was used for statistical analysis. The odds ratio (OR) and corresponding 95% confidence interval (CI) for the dichotomous variables were combined, and the mean differences (MD) and corresponding 95% CI for continuous variables were combined to obtain the combined effect of each study result. The heterogeneity among studies was evaluated using the $I^2$ and Q statistics. When there was no or less heterogeneity among studies (P $\geq$0.1 or $I^2$ <50%), a fixed effect model was selected. When there was heterogeneity among studies (P <0.1 or $I^2$ $\geq$50%), a random effect model was selected. The possible sources of heterogeneity were further analysed using subgroup and sensitivity analyses.

**Table 1. The main characteristics of included studies.**

| Author | Year | Follow-up (yr) | Age | Male (%) | Intervention | Size | Outcomes | | | | |
|---|---|---|---|---|---|---|---|---|---|---|---|
| | | | | | | | Recurrence | Progression | Side effect | Bladder irritation sign | Gross haematuria |
| RENZO COLOMBO [13] | 1996 | 3.1 | 64.3 | 84.62 | MMC+RITE/ MMC | 52 | 8/9 | | | | |
| M. BRAUSI [14] | 1998 | 2 | 70 | 75.00 | MMC+EMDA/ MMC | 28 | 2/3 | | 2/1 | 2/0 | 0/1 |
| Renzo Colombo [15] | 2003 | 2 | NA | 83.13 | MMC+RITE/ MMC | 75 | 6/23 | 2/3 | 34/21 | | 3/2 |
| SAVINO M. DI STASI [16] | 2003 | 3.6 | 66.5 | 72.22 | MMC+EMDA/ MMC | 72 | 19/27 | 3/4 | | | 8/6 |
| Wang B [17] | 2012 | 1.5 | 59.4 | 88.89 | MMC+CHT/ MMC | 18 | 3/7 | | | | |
| Gao F [18] | 2015 | 3 | 55.7 | 62.50 | MMC+CHT/ MMC | 64 | 3/10 | | 6/7 | 6/7 | |
| Liu MD [19] | 2015 | 3 | 51.5 | 57.50 | MMC+CHT/ MMC | 40 | 2/7 | | 14/10 | 10/8 | 4/2 |
| Li B [20] | 2016 | NM | 59.4 | 88.89 | MMC+CHT/ MMC | 90 | 15/35 | | 10/9 | 10/7 | |
| Ba, M. [21] | 2017 | 3.9 | 51.2 | 92.45 | MMC+CHT/ MMC | 53 | 3/7 | | | | |
| Liu QY [22] | 2017 | 6.3 | 62.6 | 83.68 | MMC+CHT/ MMC | 288 | 33/58 | | 38/46 | 25/33 | 6/7 |
| Liu XL [23] | 2017 | 3 | 39.2 | 66.18 | MMC+CHT/ MMC | 68 | 2/9 | 1/9 | | | |
| Wang X [24] | 2017 | 1.5 | 53.5 | 54.08 | MMC+LRH/ MMC | 98 | 7/12 | | | 18/17 | 8/9 |
| Su XY [25] | 2019 | 3 | 50.3 | 69.74 | MMC+CHT/ MMC | 76 | 4/16 | | 2/11 | | |
| Zhao ZH [26] | 2019 | 2 | 66.5 | 51.00 | MMC+LRH/ MMC | 100 | 5/15 | 0/3 | | | |
| Alejandro Sousa [27] | 2020 | 4.5 | NM | NM | MMC+CHT/ MMC | 68 | 7/13 | 1/4 | 21/17 | | |

NM: not mentioned; RITE: radiofrequency induced thermochemotherapy; EMDA: electromotive drug administration; CHT: conductive thermochemical therapy; LRH: loco-regional hyperthermia; MMC: mitomycin C.

## Result

### Search results and study characteristics

In total, 1,305 studies that met the relevant search criteria were retrieved from the above databases, and the available literature data were managed using Endnote. First, 497 duplicate studies were deleted, and then, 100 systematic reviews or animal studies were excluded. From the remaining 708 articles, we excluded 659 studies that were deemed irrelevant based on their titles and abstracts. The remaining 49 articles were selected and examined. Finally, 15 studies [13–27] were included in the meta-analysis. The article selection procedures are summarized in **Fig 1**. A total of 1,190 patients across 30 arm tests were included in the quantitative evaluation. **Table 1** summarizes the interventions and the effects of the results.

### Quality assessment

The quality assessment tool is shown in **Fig 2**, and the results are summarized in **Fig 3**. Among them, 1 study achieved high-quality evaluation, 11 reached medium-quality evaluation, and 3

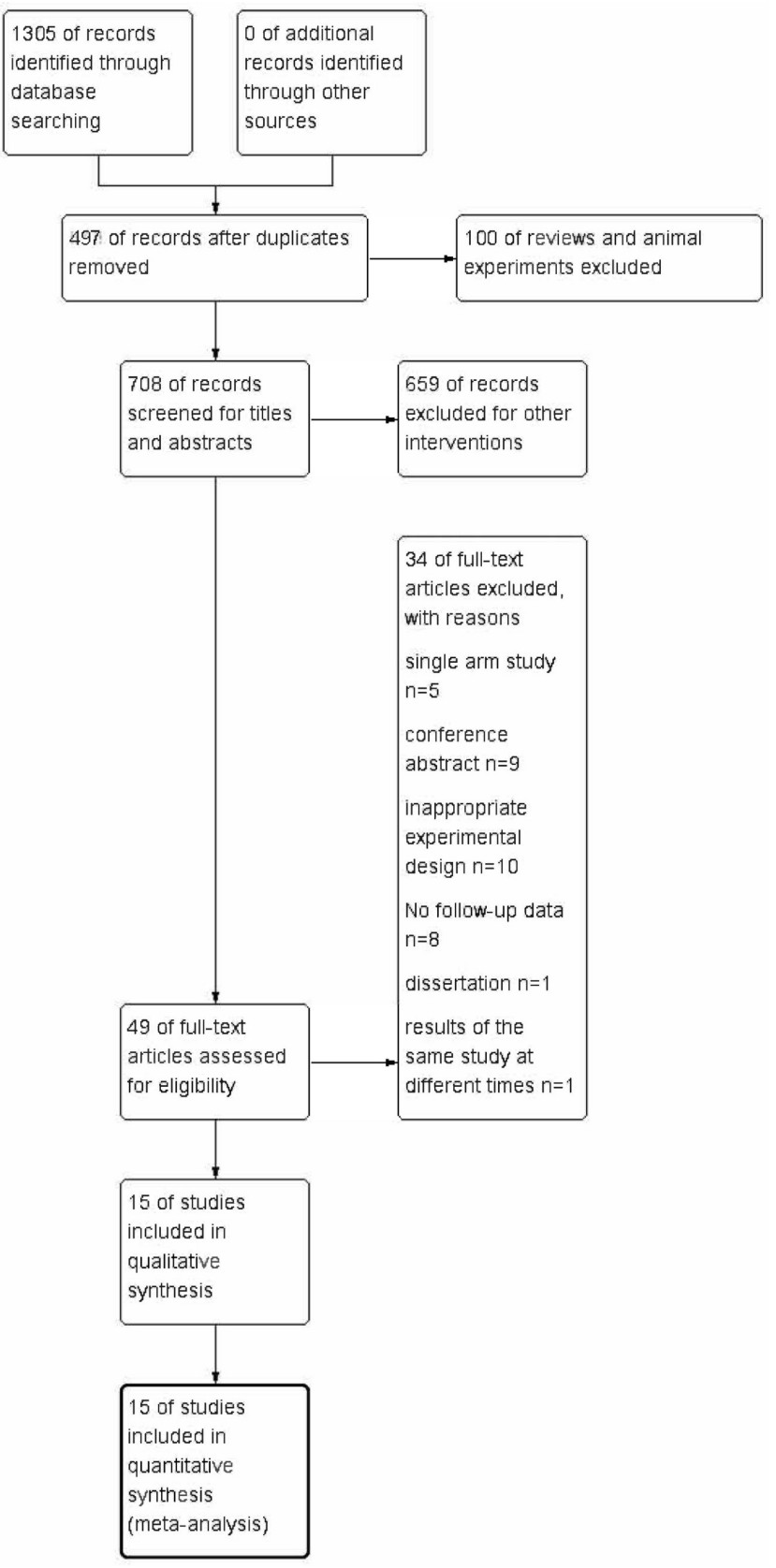

**Fig 1. Flowchart of study selection.**

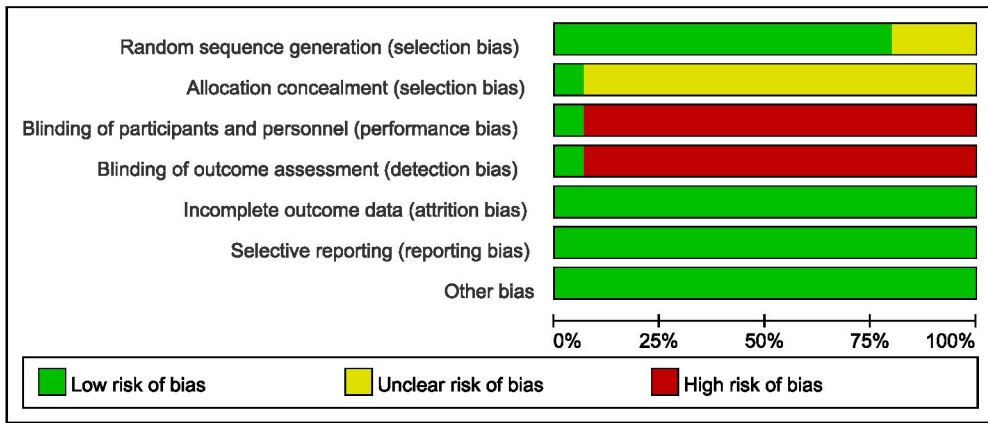

**Fig 2. Risk of bias graph.**

achieved lower-quality evaluation. On the other hand, the included studies had lower consumptive (incomplete outcome data), reporting (selective reporting), selection (random sequence generation), and other biases. However, there was a high risk in the blind method of random allocation concealment, participants and personnel, and the evaluation of results. Due to the particularity of clinical research, it was difficult to implement the blind method, and the high risk of the blind method was inevitable. The above data could be included in this meta-analysis study.

## Recurrence rate

The outcome index was included in 15 studies, and the statistical heterogeneity among the results was low (P = 0.29, $I^2$ = 14%). The results of the fixed effect model analysis (OR = 0.32, 95%CI [0.24, 0.42], P <0.00001) (**Fig 4**) showed that equipment-assisted MMC intravesical instillation was better than simple MMC instillation therapy, and the difference between the two groups was statistically significant. We also performed a subgroup analysis according to different equipments, which could be divided into four groups: MMC+EMDA/MMC, MMC+RITE/MMC, MMC+CHT/MMC, and MMC+LRH/MMC. The results showed that there was no significant statistical heterogeneity in each group. A fixed effect model was used for meta-analysis. The results of each subgroup showed that MMC perfusion therapy assisted by each of these different equipment was better than simple MMC perfusion therapy, and there were statistical differences.

## Progression rate

The outcome index was included in five studies, and the statistical heterogeneity among the results was low (P = 0.48, $I^2$ = 0%). The results of the fixed effect model analysis (OR = 0.29, 95% CI [0.12, 0.67], P = 0.004) (**Fig 5**) showed that equipment-assisted MMC intravesical instillation was better than simple MMC instillation therapy, and the difference between the two groups was statistically significant. We also performed a subgroup analysis according to variations in equipment, which could be divided into four groups: MMC+EMDA/MMC, MMC+RITE/MMC, MMC+CHT/MMC, and MMC+LRH/MMC. The results of the subgroup analysis showed that there was no significant statistical heterogeneity in each group. A fixed effect model was used for meta-analysis. The results of each subgroup showed that MMC

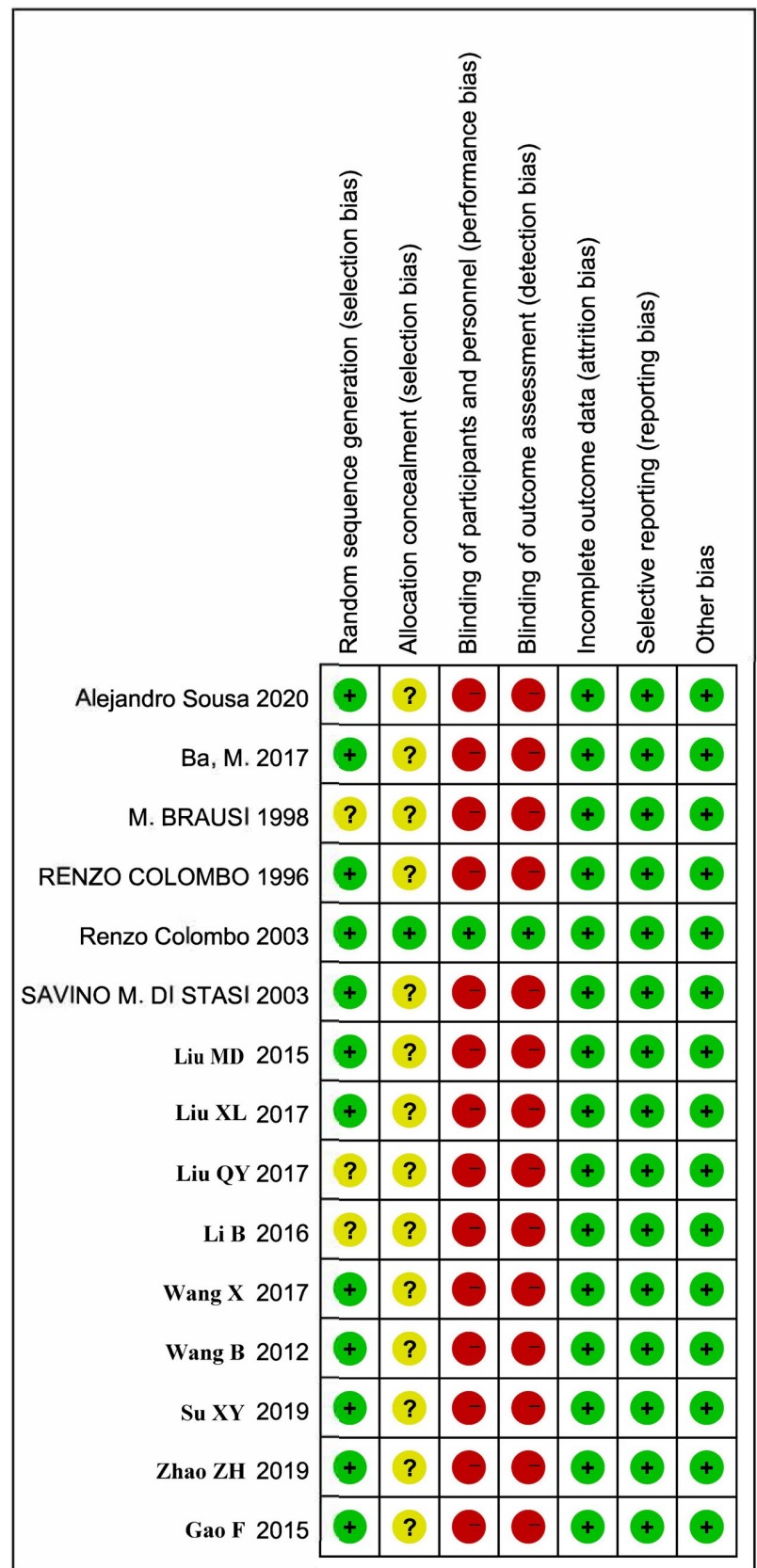

**Fig 3. Risk of bias summary.**

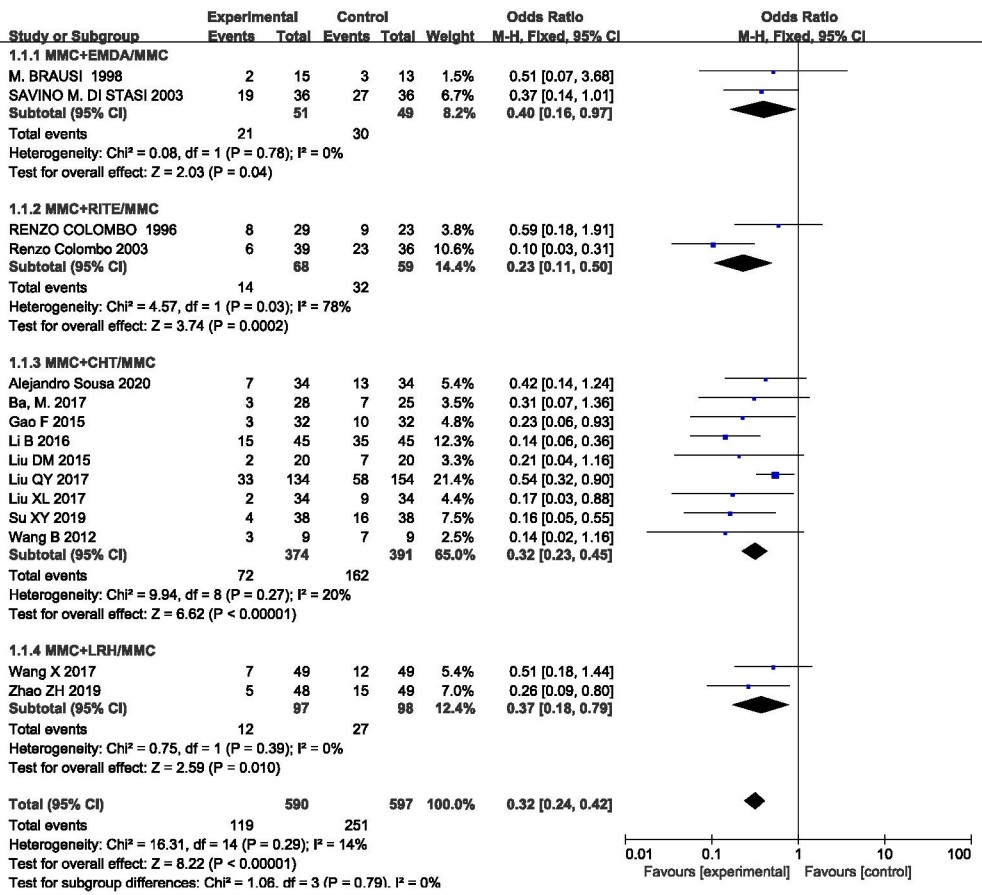

**Fig 4. Forest plots for the meta-analysis of the recurrence rates by the different equipment.**

perfusion therapy assisted by different equipment was better than simple MMC perfusion therapy, However, the difference was statistically different only in the MMC+CHT/MMC subgroup.

## Side-effect profiles

This outcome was assessed in eight studies, and the statistical heterogeneity among the results was slightly larger (P = 0.03, $I^2$ = 55%). The results were analysed using a random effect model (OR = 1.22, 95%CI [0.68,2.20], P = 0.50) (**Fig 6**), which showed that MMC infusion alone was better than equipment-assisted MMC bladder instillation, but there was no significant difference between the approaches. In order to explore heterogeneity, we carried out a subgroup analysis according to different equipment, which could be divided into three groups: MMC +EMDA/MMC, MMC+RITE/MMC, and MMC+CHT/MMC. There was statistical heterogeneity among the groups. A random effect model was used for meta-analysis. The results suggested that different device-assisted intravesical installation therapies may have been the source of the heterogeneity. Sensitivity analysis was then used to further explore the source of the heterogeneity. After excluding the study by Su [25], the heterogeneity decreased significantly (P = 0.21, $I^2$ = 28%), and the combined total effect changed (OR = 1.42, 95%CI [0.90, 2.26], P = 0.13). However, there was no significant change in the results, which were still not statistically significant, indicating that the results of this study were reliable.

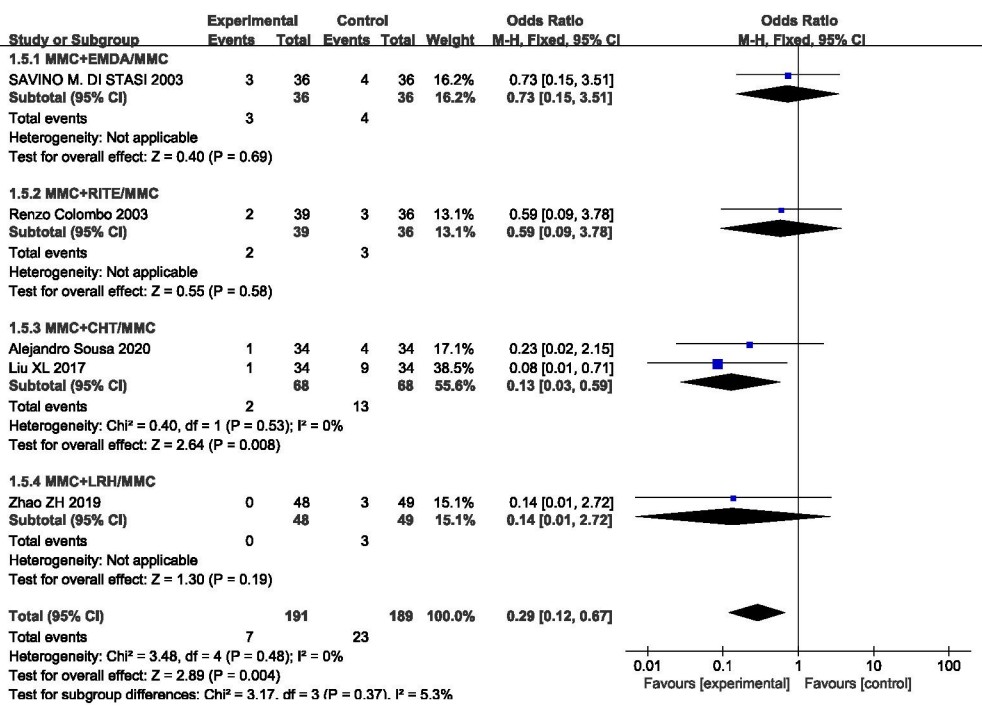

**Fig 5. Forest plots for the meta-analysis of the progression rates by the different equipment.**

## Bladder irritation

Six studies included this outcome, and there was no statistical heterogeneity among the results (P = 0.78, $I^2$ = 0%). The results were analyzed using a fixed effect model (OR = 1.06, 95%CI

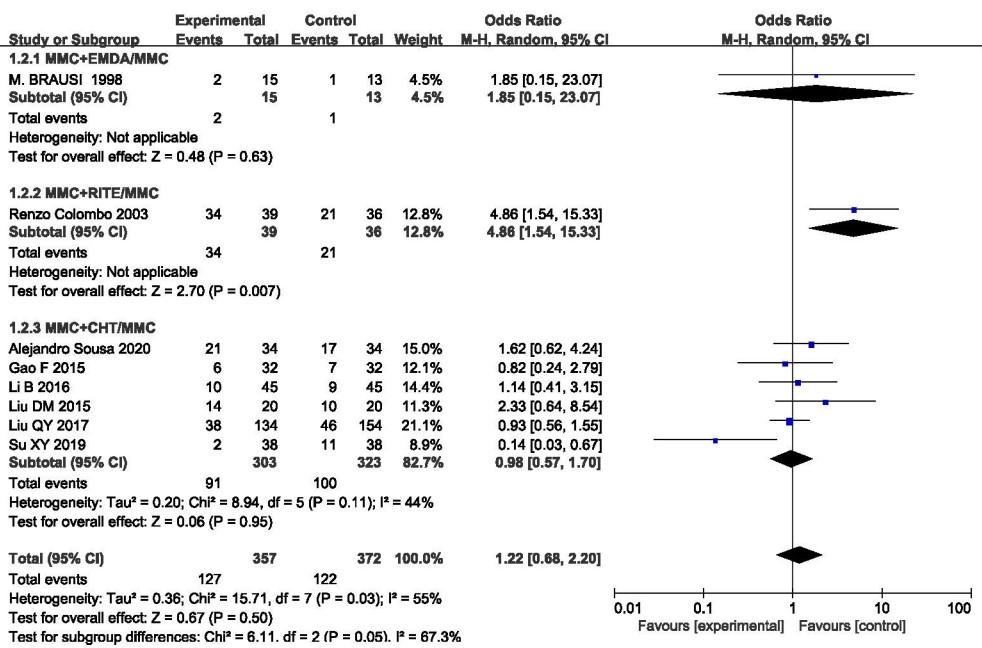

**Fig 6. Forest plots for the meta-analysis of the side-effect profiles by the different equipment.**

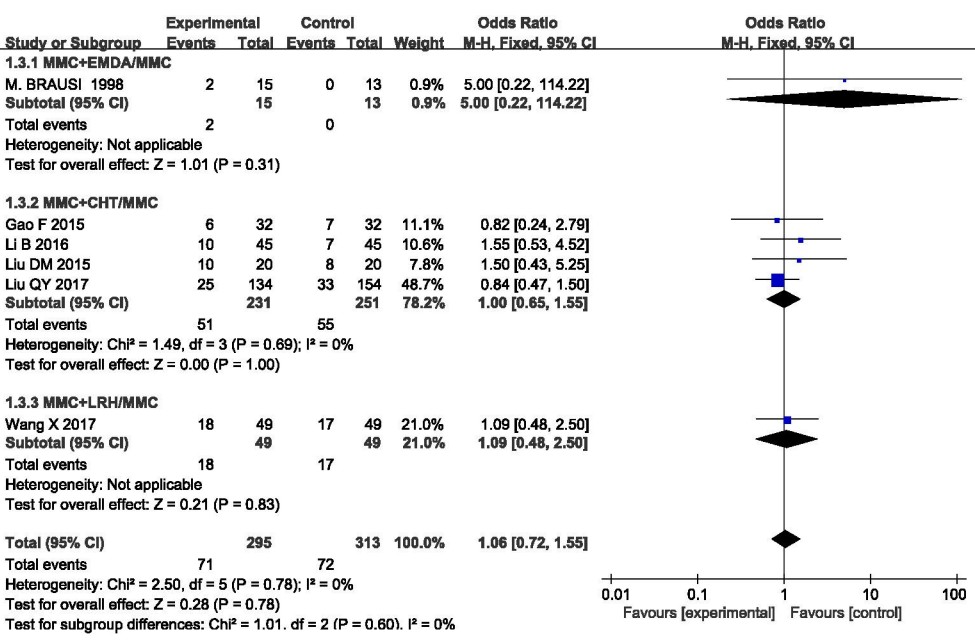

**Fig 7. Forest plots for the meta-analysis of the bladder irritation by the different equipment.**

[0.72, 1.55], P = 0.78) (Fig 7), indicating that MMC infusion alone was better than equipment-assisted MMC bladder instillation, but there was no significant difference. We also carried out subgroup analysis according to different equipment, which could be divided into three groups: MMC+EMDA/MMC, MMC+CHT/MMC, and MMC+LRH/MMC. There was no statistical heterogeneity among the groups. A fixed effect model was used for meta-analysis. The results of each subgroup showed that simple MMC perfusion therapy was better than equipment-assisted MMC perfusion therapy, but there was no statistical difference.

## Gross haematuria

This outcome index was included in six studies, and the statistical heterogeneity among the results was low (P = 0.88, $I^2$ = 0%). A fixed effect model was used to analyse the results (OR = 1.11, 95%CI [0.64, 1.94], P = 0.72) (Fig 8). MMC infusion alone was better than equipment-assisted MMC bladder instillation, but there was no significant difference. We also carried out subgroup analysis according to different equipment, which could be divided into four groups: MMC+EMDA/MMC, MMC+RITE/MMC, MMC+CHT/MMC, and MMC+LRH/MMC. There was no statistical heterogeneity among the groups. A fixed effect model was used for meta-analysis. The results of each subgroup except for the MMC+LRH/MMC group suggested that MMC perfusion alone is better than equipment-assisted MMC instillation therapy, but there is no statistical difference.

## Publication bias analysis

In this study, the funnel plot was used to detect whether there was publication bias (Fig 9). The results showed that except for the recurrence rate and progression rate outcomes, the other funnel plots were roughly symmetrically distributed. This means that the studies related to the recurrence rate and progression rate outcome indices have a publication bias to some extent, and the other studies included in the outcome index have a low publication bias.

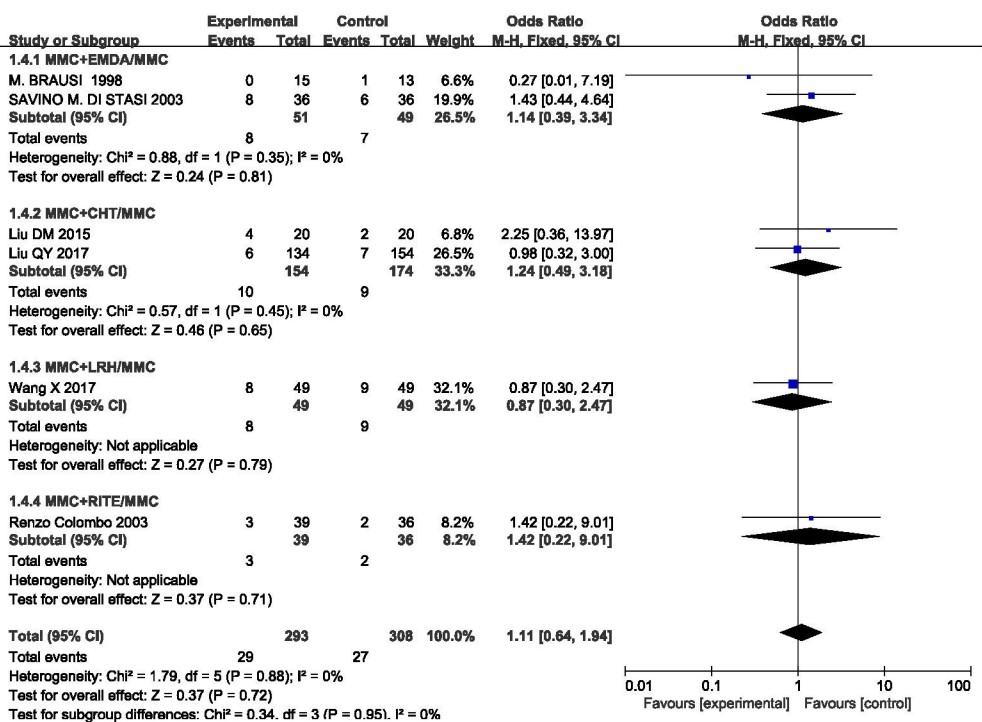

**Fig 8. Forest plots for the meta-analysis of the gross haematuria by the different equipment.**

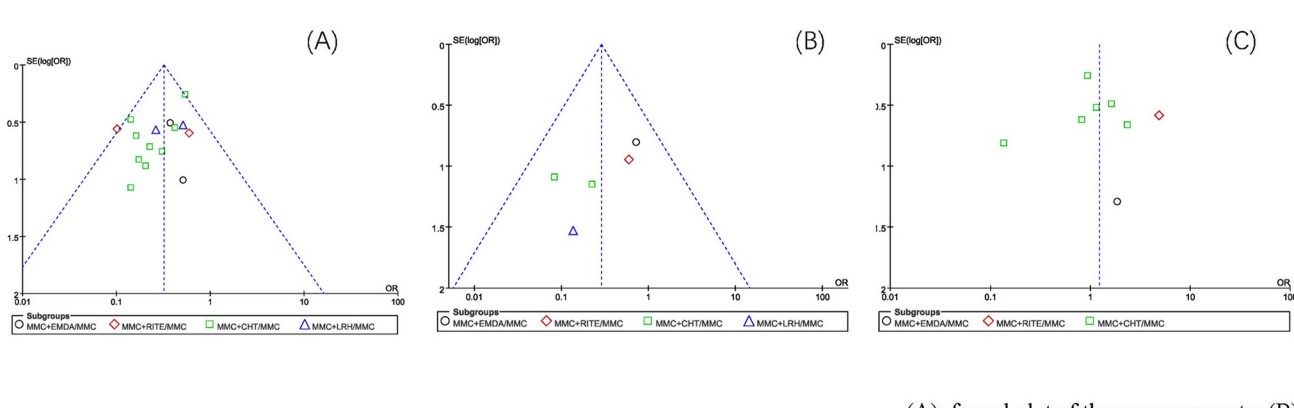

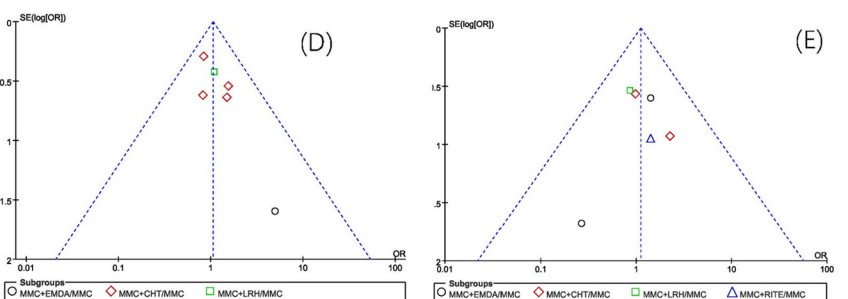

(A): funnel plot of the recurrence rate; (B): funnel plot of the progression rate; (C): funnel plot of the side-effect profiles; (D): funnel plot of the bladder irritant; (E): funnel plot of the gross haematuria;
RITE: radiofrequency induced thermochemotherapy; EMDA: electromotive drug administration; CHT: conductive thermochemical therapy; LRH: loco-regional hyperthermia; MMC: mitomycin C.

**Fig 9. Funnel plots for the publication bias of the included studies based on different outcome indicators.**

## Discussion

Bladder cancer-related diseases have the 13th highest mortality rate among all diseases [1]. This finding can be explained partly by the generally low survival rate of muscular-invasive bladder cancer: the 5-year survival rate is 29–57% [28, 29]. Improving the postoperative intravesical instillation efficiency of patients with NMIBC can effectively prevent the progression of muscular-invasive bladder cancer. Based on this observation, our study systematically evaluated the clinical efficacy and safety of MMC intravesical instillation in patients with NMIBC. The outcomes of interest included recurrence rates, progression rates, side-effects, gross haematuria, and bladder irritation. The results showed that equipment-assisted MMC intravesical instillation therapy could ensure safety and significantly reduce the recurrence and progression rates.

This study included equipment-assisted intravesical MMC instillation, which aims to optimise the established treatment plan and enhance the efficacy of chemotherapy. LRH mainly transfers energy through focused electromagnetic waves in vitro to increase tissue temperature. This method is mainly performed using the BSD-2000V hyperthermia and AMC 70 MHz systems [30]. CHT is achieved by conducting heat transfer from a heated circulating liquid. This method is mainly performed using the bladder recirculation system, mild cryogenic system, and the BR-TRG-I high precision hyperthermic intraperitoneal perfusion system [11]. RITE is performed by local high temperature induced via radiofrequency in the bladder. All three approaches directly affect neoplastic cells, promote tumour angiogenesis, and trigger the immune response through hyperthermia [31]. In vitro experiments have shown that hyperthermia can cause the deformation of the cytoplasmic structure and various enzyme proteins, and induce cellular apoptosis and necrosis [32–34]. This increase in temperature increases cell membrane permeability and leads to increased drug absorption [35–37]. Hyperthermia also leads to the release of heat shock proteins (HSPs), especially HSP70, stimulates adaptive T-cell responses, and induces innate and adaptive immune responses [38, 39]. EMDA uses electrokinetic phenomena to accelerate the movement of drugs through the biological barrier (i.e., the urinary tract epithelium) by applying electric current to increase local drug concentration [40]. In vitro experiments have shown that EMDA-assisted intravesical instillation can significantly increase drug concentrations in the urinary tract epithelium, lamina propria, and muscular layer of the bladder compared to passive diffusion. Indeed, EMDA increases MMC diffusion by 4–7 times [41].

Our meta-analysis showed that equipment-assisted MMC intravesical instillation was significantly better than simple MMC intravesical instillation in preventing recurrence, irrespective of the subgroup or the combined total effect. Subgroup analysis revealed that the combined effect (OR = 0.40) in the EMDA+MMC/MMC group was the highest, followed by the LRH+MMC/MMC group (OR = 0.37), CHT+MMC/MMC group (OR = 0.32), and RITE +MMC/MMC group (OR = 0.23). We can boldly speculate that MMC intravesical instillation under RITE equipment is the best in preventing the recurrence of NMIBC. Still, more related studies are needed to verify it. Some studies have reported that MMC perfusion therapy assisted by RITE may be effective in patients with failed BCG treatment [42]. It may also be a worthy alternative for conservative treatment of refractory CIS (carcinoma in situ) patients [43]. However, from the sample size included, the CHT+MMC/MMC group analysis was more reliable, and MMC intravesical instillation under CHT was more efficient. Some studies have shown that MMC intravesical instillation under CHT may be an alternative to postpone the need for radical resection in some patients with a slow response to BCG treatment [44]. At the same time, studies have shown that MMC intravesical instillation under CHT prevents recurrence in patients with non-grade 3 BCG recurrence, and 41% of the patients remained disease-

free during a median follow-up period of 41 months [45]. In the present analysis, although the EMDA+MMC/MMC group obtained a relatively high OR value, other studies showed the advantages of EMDA device-assisted perfusion. For example, the median time of recurrence in patients after EMDA before TURBT was significantly prolonged up to 57 months [46]. Alternatively, combining BCG with EMDA+MMC is a more reliable treatment than EMDA +MMC alone, being associated with lower rates of progression, overall mortality, and comorbidity [47, 48].

In addition to this, in terms of tumour progression rates, a review outlined that progression rates under RITE device-adjuvant therapy ranged from 0–19%, with progression mostly occurring in high-risk patients that had failed bladder therapy and multiple recurrences [49]. Progression rates with CHT device-assisted therapy ranged from 0–8% [50], and one long-term follow-up study also showed lower rates of progression with CHT + MMC than with MMC alone [51]. At present, there have been no reports on the rate of progression for EMDA and LRH related research. However, this study showed that equipment-assisted intravesical instillation of MMC has significant advantages. Notably, only CHT+MMC significantly reduces the tumour progression rate; other equipment-assisted MMC instillation had no significant advantage over MMC intravesical instillation alone. In spite of this, the aggregate effect results were consistent with the results of each subgroup, which showed that the results of this analysis were relatively robust. Considering the small number of studies and samples included in this outcome index, it is reasonable to believe that increasing the overall sample size of the study will highlight the significant advantage of MMC bladder instillation assisted by corresponding equipment in reducing tumour progression.

On the side-effect outcome index of this research, the results of this study showed no significant change in the overall effect by excluding Su [25], so the study was retained in this meta-analysis. However, in the CHT+MMC/MMC subgroup, the heterogeneity decreased significantly after removal (P = 0.64, $I^2$ = 0%), and the total effect was combined using fixed effect model analysis (OR = 1.11, 95%CI [0.77,1.61], P = 0.58), which was different from the previous analysis (OR = 0.98, 95%CI [0.57, 1.70], P = 0.95). The results changed, but there was still no statistically significant difference. However, more studies are needed to confirm whether intravesical instillation assisted by CHT equipment increases this risk, and whether there is a significant difference compared to other approaches. Through subgroup analysis, we also found that the RITE+MMC/MMC group analysis showed that MMC perfusion under RITE equipment could significantly produce more side-effects (OR = 4.86, 95%CI [1.54, 15.33], P = 0.007). At the same time, some studies have suggested that because RITE is prone to serious side-effects, it is only suitable for specific recurrent NMIBC patients [52]. However, there is little other evidence about this type of research, and the reliability of the results needs to be verified by more research. The inclusion of side-effects in this study was not sufficient, and our qualitative analysis results were insufficient. However, some studies systematically summarized the safety of related equipment. The main adverse events of patients using CHT were noninfective cystitis (37.2%), suprapubic pain (23.3%), and bladder spasms or urination (22.1%). The most common adverse events in patients treated with RITE were cystitis or cumulative lower urinary tract symptoms (LUTS) (32%), suprapubic pain (30%), and bladder spasm (20%). The most common adverse reactions after treatment were cumulative lower urine, such as urinary incontinence (12.6%), nocturnal urine (23.9%) and haematuria (17.2%). In total, 3.3% of the patients had urethral stricture, but most of the adverse events were self-limited and transient [6]; Patients using EMDA terminated treatment prematurely because of LUTS, haematuria, and intolerable catheters [48]. Few studies have examined LRH, and more research is needed to understand its comparative safety.

In terms of bladder irritation and frank haematuria, the analysis showed that there was no significant difference between device-assisted MMC intravesical instillation and MMC alone. However, in the subgroup analysis, we found that the incidence of frank haematuria in MMC intravesical instillation with CHT equipment (OR = 1.00) was lower than that with LRH (OR = 1.09) or EMDA equipment (OR = 5). In terms of gross haematuria, the incidence of MMC bladder instillation (OR = 0.87) with LRH was lower than that with EMDA (OR = 1.14), CHT (OR = 1.24), and RITE (OR = 1.42). However, limited to the inclusion of a small number of research samples, the above results need to be verified with more data.

However, there were some limitations in our whole analysis. First, moderate bias risk studies accounted for most of the included RCTs, which reduces the reliability of the ranking evidence of the analysis results. The unspecified random process and non-blind research design were the main reasons for the low quality of this RCT. Because of the ethics of clinical trial design, it was difficult to achieve a completely blind method. Secondly, in the subgroup analysis based on different outcome indicators and according to different equipment, due to the small sample size of some studies, the reliability of other analysis results except CHT equipment needed to be proved by more other studies. Moreover, each type of equipment-assisted perfusion therapy could not be guaranteed to be based on the same device, and could not reveal the impact of potential factors on the results of the study. However, the results of this study showed that heterogeneity was acceptable, which fully showed the rationality of the feasible combined analysis of each study. Finally, some outcome indicators had a publication bias, which affected the credibility of the results of this study to some extent.

In summary, the present study was the first meta-analysis based on adjuvant device bladder perfusion with MMC that confirms device-assisted MMC bladder perfusion significantly reduces rates of tumour recurrence and progression in patients with NMIBC after TURBT relative to single MMC bladder perfusion with comparable safety. Through subgroup analysis, the efficacy and safety of perfusion therapy with different devices were further investigated, providing prospective ideas for subsequent studies comparing various devices. This study was only a small step on the road to explore the best treatment options. In order to optimize the treatment of patients with NMIBC, more prospective, multicenter, large sample RCTs are needed to provide more substantial evidence. For example, a positive study could be carried out to compare the benefits between two specific system devices; to explore the benefits of different perfusion treatment temperatures, different times and intervals of treatment, and different populations of patients with NMIBC; to compare the efficacy of equipment-assisted MMC infusion with other drug combinations or continuous treatments; and the benefits of equipment-assisted therapy for patients who fail or relapse after other drug treatments. The high cost of RITE therapy, followed by EMDA and CHT [6], may limit its promotion and result in less related research, resulting in a vicious cycle. In other words, clinicians need to comprehensively consider the cost and safety benefits while reducing the tumour recurrence and progression rates and ensuring safety. We believe this study may provide some degree of guidance for further clinical research.

## Supporting information

**S1 File. PRISMA 2020 checklist.**
(DOCX)

**S2 File. The detailed search processes.**
(PDF)

## Author Contributions

**Conceptualization:** Weijian Zhou.

**Data curation:** Weijian Zhou, Changying Hu.

**Formal analysis:** Weijian Zhou.

**Investigation:** Weijian Zhou, Jianping Liu.

**Methodology:** Weijian Zhou, Jianping Liu, Dongdong Mao.

**Project administration:** Weijian Zhou.

**Resources:** Weijian Zhou, Dianjun Gao.

**Software:** Weijian Zhou, Jianping Liu, Dianjun Gao.

**Supervision:** Weijian Zhou, Dongdong Mao, Dianjun Gao.

**Validation:** Weijian Zhou, Dongdong Mao, Changying Hu, Dianjun Gao.

**Visualization:** Weijian Zhou, Changying Hu, Dianjun Gao.

**Writing – original draft:** Weijian Zhou.

**Writing – review & editing:** Weijian Zhou, Dianjun Gao.

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
