## [Decision Letter · Decision Letter 0]

22 Sep 2022

PONE-D-22-24475The Clinical Efficacy and Safety of Equipment-Assisted Intravesical Instillation of Mitomycin C After Transurethral Resection of Bladder Tumor in Patients with Non-Muscular Invasive Bladder Cancer: A Meta-AnalysisPLOS ONE

Dear Dr. Zhou,

Thank you for submitting your manuscript to PLOS ONE. After careful consideration, we feel that it has merit but does not fully meet PLOS ONE’s publication criteria as it currently stands. Therefore, we invite you to submit a revised version of the manuscript that addresses the points raised during the review process.

A marked-up copy of your manuscript that highlights changes made to the original version. You should upload this as a separate file labeled 'Revised Manuscript with Track Changes'.An unmarked version of your revised paper without tracked changes. You should upload this as a separate file labeled 'Manuscript'.

We look forward to receiving your revised manuscript.

Kind regards,

Alessandro Rizzo

Academic Editor

PLOS ONE

Journal Requirements:

Additional Editor Comments:

Please note that if a reviewer has requested citations to specific articles, those articles should only be cited if they are directly relevant to the study. If you find that any number of the requested citations are irrelevant or inappropriate, please state this in your Response to Reviewers for each article you assess to be irrelevant or inappropriate to cite.

Reviewers' comments:

Reviewer's Responses to Questions

**Comments to the Author**

1. Is the manuscript technically sound, and do the data support the conclusions?

Reviewer #1: Partly

Reviewer #2: Yes

Reviewer #3: Yes

2. Has the statistical analysis been performed appropriately and rigorously? 

Reviewer #1: Yes

Reviewer #2: Yes

Reviewer #3: I Don't Know

3. Have the authors made all data underlying the findings in their manuscript fully available?

Reviewer #1: Yes

Reviewer #2: Yes

Reviewer #3: Yes

4. Is the manuscript presented in an intelligible fashion and written in standard English?

Reviewer #1: No

Reviewer #2: Yes

Reviewer #3: Yes

5. Review Comments to the Author

Reviewer #1: Dear Editor, thank you so much for inviting me to revise this manuscript about bladder cancer.

This study addresses a current topic.

The manuscript is quite well written and organized. English should be improved.

Figures and tables are comprehensive and clear.

The introduction explains in a clear and coherent manner the background of this study.

We suggest the following modifications:

• Introduction section: although the authors correctly included important papers in this setting, we believe the systemic treatment scenario for bladder cancer patients should be better discussed in the introduction section and some recently published papers added ( PMID: 35048736 ; PMID: 33516645 ), only for a matter of consistency. We think it might be useful to introduce the topic of this interesting study.

• Methods and Statistical Analysis: nothing to add.

• Discussion section: Very interesting and timely discussion. Of note, the authors should expand the Discussion section, including a more personal perspective to reflect on. For example, they could answer the following questions – in order to facilitate the understanding of this complex topic to readers: what potential does this study hold? What are the knowledge gaps and how do researchers tackle them? How do you see this area unfolding in the next 5 years? We think it would be extremely interesting for the readers.

However, we think the authors should be acknowledged for their work. In fact, they correctly addressed an important topic, the methods sound good and their discussion is well balanced.

One additional little flaw: the authors could better explain the limitations of their work, in the last part of the Discussion.

We believe this article is suitable for publication in the journal although major revisions are needed. The main strengths of this paper are that it addresses an interesting and very timely question and provides a clear answer, with some limitations.

We suggest a linguistic revision and the addition of some references for a matter of consistency. Moreover, the authors should better clarify some points.

Reviewer #2: The purpose of this study is to identify the clinical efficacy and safety of instrument assisted intravesical instillation of mitomycin C in patients with non muscle invasive bladder cancer after TURBT. The data exclusion and inclusion are reasonable, and the results are reliable and can guide clinical practice. However, similar studies seemed to be published and this study was not creative.

Reviewer #3: The authors presented the meta-analysis showing that equipment-assisted MMC intravesical instillation therapy could ensure safety and significantly reduce the recurrence rate. Cancer progression to muscle invasive cancer is also an important endpoint. Progression rate should be included as endpoint in their meta-analysis.

6. PLOS authors have the option to publish the peer review history of their article (what does this mean?). If published, this will include your full peer review and any attached files.

Reviewer #1: No

Reviewer #2: No

Reviewer #3: No

---

## [Author Response · Author response to Decision Letter 0]

5 Oct 2022

Dear editor and reviewers,

We feel great thanks for your professional review work on our article. Thanks to the reviewers and editors for your suggestions. As you are concerned, there are several problems that need to be explained and addressed. According to your comments, we have made some corrections to our previous draft, the detailed corrections and explanation are listed below.

Comment 1. Is the manuscript technically sound, and do the data support the conclusions? The manuscript must describe a technically sound piece of scientific research with data that supports the conclusions. Experiments must have been conducted rigorously, with appropriate controls, replication, and sample sizes. The conclusions must be drawn appropriately based on the data presented. Reviewer #1: Partly. Reviewer #2: Yes. Reviewer #3: Yes

Response 1: We thank the reviewers for their comments, and for the response given by reviewer 1, he has followed up with an explanation of exactly why he gave ‘partly’. We have revised and elaborated on the specific comments related to reviewer 1 below, which we hope will change your opinion. Thanks again to reviewers 2 and 3 for their affirmation.

Comment 2. Has the statistical analysis been performed appropriately and rigorously? Reviewer #1: Yes. Reviewer #2: Yes. Reviewer #3: I Don't Know

Response 2: Thanks to the reviewers' comments, this article was verified by Suzhen Wang, a professor and doctoral supervisor of epidemiology and health statistics in the School of Public Health of Weifang Medical University. The following is her relevant information: Email: wangsz@wfmc.edu.cn; Phone: +8613964739059; Address: WeiFang Medical University. 4948 Shengli East Street, Weifang City, Shandong Province, China. Hopefully, this will allay the concerns of reviewers.

Comment3. Have the authors made all data underlying the findings in their manuscript fully available? Reviewer #1: Yes. Reviewer #2: Yes. Reviewer #3: Yes

Response 3: We thank the reviewers for their endorsement.

Comment 4. Is the manuscript presented in an intelligible fashion and written in standard English? Reviewer #1: No. Reviewer #2: Yes. Reviewer #3: Yes

Response 4: Thank you for your significant reminding. According to your suggestion, we corrected the grammatical errors and made an effort to polish the whole manuscript. These changes will not influence the content and framework of the paper. We would like to confirm that the suitably revised manuscript is understandable to readers. The proof of retouching has been uploaded to the supplementary document.

Reviewer #1

Comment 1: Introduction section: although the authors correctly included important papers in this setting, we believe the systemic treatment scenario for bladder cancer patients should be better discussed in the introduction section and some recently published papers added ( PMID: 35048736 ; PMID: 33516645 ), only for a matter of consistency. We think it might be useful to introduce the topic of this interesting study.

Response: We would like to thank the reviewers for their suggestions. We have read the full text of the articles you recommended and found that the two articles are valuable studies and still have some relevance to the topic of this study. Both articles have now been cited in the introduction section of the original article. This addition makes my article more likely to be attractive.

Comment 2: Discussion section: Very interesting and timely discussion. Of note, the authors should expand the Discussion section, including a more personal perspective to reflect on. For example, they could answer the following questions – in order to facilitate the understanding of this complex topic to readers: what potential does this study hold? What are the knowledge gaps and how do researchers tackle them? How do you see this area unfolding in the next 5 years? We think it would be extremely interesting for the readers.

Response: Many thanks to the reviewer for his careful advice. As you said, the original article was indeed missing this part of the description. We have added it in the last paragraph of the discussion, and by adding it, the article discussion looks more fleshed out. Thank you again for your suggestion.

Comment 3: the authors could better explain the limitations of their work, in the last part of the Discussion.

Response: We thank the reviewers for their suggestions. We have clarified the limitations of this study in the penultimate paragraph of the Discussion section. Regarding the word ‘limitations’ that appeared in the last paragraph of the discussion in the previous version of the manuscript, it’s my reference to the content of the penultimate paragraph and may have caused you a misunderstanding here. But thank you very much for your reminder, and we have adjusted the content of the last paragraph of the discussion section of the article in the hope that it can be better understood by you and the readers.

Reviewer #2

Comment: The purpose of this study is to identify the clinical efficacy and safety of instrument assisted intravesical instillation of mitomycin C in patients with non muscle invasive bladder cancer after TURBT. The data exclusion and inclusion are reasonable, and the results are reliable and can guide clinical practice. However, similar studies seemed to be published and this study was not creative.

Response: We thank the reviewers for their comments. We believe that our study is innovative in demonstrating that device-assisted bladder instillation with MMC significantly reduces recurrence and progression of TURBT patients with NMIBC and improves their quality of life by meta-analysis. First of all, this study is the first meta-analysis based on bladder instillation MMC with each adjunct device and systematically evaluates the effectiveness and safety of bladder instillation with device-assisted MMC. The overall heterogeneity of the combined effect (P=0.29, I2=14%) demonstrates the rationality and feasibility of combining across studies. Second, although previous studies have demonstrated the superiority of bladder instillation MMC with an assisted device, they were based on a single small sample. We pooled the studies through meta-analysis to increase the sample size of the trial, which also confirmed the above results and improved the level of evidence-based medicine, making the study results more reliable and more stable and reliable to serve clinical work. Finally, we further explored the efficacy and safety of perfusion therapy with different devices through subgroup analysis, which to some extent provides clinicians with multiple selective options for treatment and provides prospective ideas for subsequent studies comparing various devices. We again thank the reviewers for this comment and realize that these advantages may not have been expressed clearly enough in the previous manuscript. We have modified the original manuscript, especially in the last part of the introduction and the last paragraph of the discussion, to make these clearer.

Reviewer #3

Comment: The authors presented the meta-analysis showing that equipment-assisted MMC intravesical instillation therapy could ensure safety and significantly reduce the recurrence rate. Cancer progression to muscle invasive cancer is also an important endpoint. Progression rate should be included as endpoint in their meta-analysis.

Response: Thank you very much for your comments. Then we read the included literature again and found that 5 articles contained the outcome indicators of progress, so through the analysis, it has been concluded that equipment-assisted MMC bladder instillation therapy can significantly reduce the tumor progression rate [OR = 0.29, 95% CI (0.12, 0.67), P=0.004]. Relevant content including results and discussions has been supplemented in the manuscript. Once again, we would like to thank the reviewers for their keen insight, which not only enriches the content of the article, but also makes we learn the rigor and comprehensiveness of scientific research in the future. We hope it will satisfy you.

We appreciate for your warm work earnestly, and hope that the correction will meet with approval. We really need your chance again. Should you have any questions, please contact us without hesitate. 

Once again, thank you very much for your comments and suggestions.

Yours Sincerely,

Weijian Zhou

---

## [Editor Report · Decision Letter 1]

7 Oct 2022

The Clinical Efficacy and Safety of Equipment-Assisted Intravesical Instillation of Mitomycin C After Transurethral Resection of Bladder Tumor in Patients with Non-Muscular Invasive Bladder Cancer: A Meta-Analysis

PONE-D-22-24475R1

Dear Dr. Zhou,

We’re pleased to inform you that your manuscript has been judged scientifically suitable for publication and will be formally accepted for publication once it meets all outstanding technical requirements.

Kind regards,

Alessandro Rizzo

Academic Editor

PLOS ONE
---

## [Editor Report · Acceptance letter]

13 Oct 2022

PONE-D-22-24475R1 

The clinical efficacy and safety of equipment-assisted intravesical instillation of mitomycin C after transurethral resection of bladder tumour in patients with nonmuscular invasive bladder cancer: a meta-analysis 

Dear Dr. Zhou:

I'm pleased to inform you that your manuscript has been deemed suitable for publication in PLOS ONE. Congratulations! Your manuscript is now with our production department. 

Kind regards, 

on behalf of

Dr. Alessandro Rizzo 

Academic Editor

PLOS ONE